# Latent Structure Matching for Knowledge Transfer in Reinforcement Learning

**Yi Zhou**  **and Fenglei Yang \***

School of Computer Engineering and Science, Shanghai University, Shanghai 200444, China;
zhouyiyizhou@shu.edu.cn
\*   Correspondence: flyang@shu.edu.cn

**Abstract:** Reinforcement learning algorithms usually require a large number of empirical samples and give rise to a slow convergence in practical applications. One solution is to introduce transfer learning: Knowledge from well-learned source tasks can be reused to reduce sample request and accelerate the learning of target tasks. However, if an unmatched source task is selected, it will slow down or even disrupt the learning procedure. Therefore, it is very important for knowledge transfer to select appropriate source tasks that have a high degree of matching with target tasks. In this paper, a novel task matching algorithm is proposed to derive the latent structures of value functions of tasks, and align the structures for similarity estimation. Through the latent structure matching, the highly-matched source tasks are selected effectively, from which knowledge is then transferred to give action advice, and improve exploration strategies of the target tasks. Experiments are conducted on the simulated navigation environment and the mountain car environment. The results illustrate the significant performance gain of the improved exploration strategy, compared with traditional $\epsilon$-greedy exploration strategy. A theoretical proof is also given to verify the improvement of the exploration strategy based on latent structure matching.

**Keywords:** latent structure matching; reinforcement learning; transfer learning; action advice; policy improvement; mountain car

## 1. Introduction

Reinforcement learning (RL) is where an agent guides its actions based on the rewards obtained from the trial-and-error interaction with the environment [1,2]. The ultimate goal of RL is to achieve a mapping from states to actions with maximum return [3]. In recent years, reinforcement learning has been successfully applied to some aspects of life such as medical care [4], finance [5], and transportation [6]. However, in RL, a small change on the problem configuration usually leads to a complete new training, which inevitably brings a slow convergence and the growing requirement of a large amount of empirical samples. The introduction of transfer learning [7] in RL provides a promising solution to this problem.

Transfer learning guides the transfer of useful knowledge from well-learned source tasks to target tasks [8]. In RL, the knowledge obtained from previous situations can be reused as heuristics to achieve effective knowledge transfer, thus speeding up the learning procedure in new situations and reducing sample request [3]; moreover, knowledge transfer is able to mitigate much the issue caused by a change on the problem configuration as mentioned above.

However, if an inappropriate source task is selected in knowledge transfer, it will lead to negative transfer [9], which will slow down or even disrupt the learning of a new target task. Therefore, it is very important for knowledge transfer to select appropriate source tasks that have a high degree of matching with target tasks.

The similarity estimation of tasks is the main way to select matched source tasks in the existing works on knowledge transfer for RL. The works proposed at the beginning are semi-automatic, because they require human intervention for similarity estimation. A typical work [10] defined a one-to-one mapping between states and actions to estimate similarity. In the subsequent work [11], a many-to-one mapping of relevant states and actions was defined. These semi-automatic works require manual parameterization of defined mappings; nevertheless, the optimal values of the parameters are not easy to reach by experience or intuition [12]. In order to overcome this problem, some mechanisms [13,14] to select optimal mappings were developed. However, human intervention is still needed in these works [12].

Later, in order to solve the problems in the earlier works mentioned above, some automatic similarity estimation works [15–21] are presented. These works made use data-driven similarity metrics, including the Markov decision process (MDP) similarity metric [20], the Hausdorff metric [21], and the Kantorovich metric [21]. These works are highly interpretative, but computationally expensive and have trouble when handling situations involving a large number of tasks [19].

Recently, clustering algorithms [22] were used in some works to tackle large number of tasks. In these works, the clustering of policies, value functions, rewards, and dynamics of tasks, were modeled as random process to estimate the similarity [23,24]. However, these clustering-based works suffer from the bias from the modeling for clustering, and are sensitive to noise and variation in policies, dynamics, and value functions of tasks, resulting in the inaccurate estimation of task similarity.

The classification and characteristics of existing similarity estimation methods mentioned above are summarized in Table 1.

**Table 1.** Classification of similarity metrics.

| Categories of Similarity Metrics | Features | Related References |
|---|---|---|
| Semi-automatic | Depend on the experience of experts | [10,11,13,14] |
| Automatic(non-clustering-based) | Highly interpretative, but computationally expensive | [15–21] |
| Automatic(clustering-based) | Can tackle large number of tasks, but sensitive to noise | [23,24] |

Classification and characteristics of existing similarity estimation methods.

To address the limitation of existing works, we propose a novel task matching algorithm, namely latent structure matching (LSM), which uses low-dimensional embedding to extract latent essential factors and relationships of tasks. Furthermore, through low-dimensional alignment, LSM effectively matches tasks in large number, and estimates their similarity, thus finding highly-matched source tasks. Based on the matched source tasks, useful exploratory advice is formed to improve exploration strategy, and to accelerate the learning of target tasks. The experiments are conducted on the simulated maze navigation environment and the mountain car environment. The maze navigation environment is an environment with discrete state and action space, and the mountain car environment is an environment with continuous state space. The results of experiments illustrate the significant performance improvement of RL agents, when taking improved exploration strategies with the action advice from the matched source tasks selected effectively by LSM.

Our main contributions in this paper can be summed up as follows:

(i)　A novel task matching algorithm, namely LSM, is proposed for RL to derive latent structures of value functions of tasks, and align the structures for similarity estimation. Structural alignment permits efficient matching of large number of tasks, and locating the correspondences of tasks.

(ii)　Based on LSM, we present an improved exploration strategy, that is built on the knowledge obtained from the highly-matched source task. This improved strategy reduces random exploration in value function space of tasks, thus effectively improving the performance of RL agents.

(iii)　A theoretical proof is given to verify the improvement of exploration strategy with the latent structure matching-based knowledge transfer (Please see Appendix B).

In the remainder of this paper, related work is first discussed in Section 2. The proposed method is introduced in Section 3. The detailed experimental procedures and experimental results are presented in Section 4. Finally, a brief conclusion is drawn in Section 5.

## 2. Related Work

### 2.1. Knowledge Transfer in RL

The sample complexity and learning inefficiency of the RL algorithms are two of the most critical aspects that obstruct its feasibility in practical applications [25]. Transfer learning is introduced to address these problems.

To perform effective knowledge transfer in RL, it is of great importance to select appropriate source tasks that are highly-matched with the target task. In most of the existing works, similarity estimation is the main way of selecting a matched source task. Early works were semi-automatic, as they conducted the similarity estimation of tasks according to human experience or intuition. Taylor and Whiteson [26] defined one-to-one inter-task mapping (TVITM) as an action in one task and its image in another task being similar with each other. The similarity was loosely defined so that it depended on how the action affected the agent's state and what reward was received. TVITM was subsequently used by Taylor and Stone [10] for knowledge transfer in RL to speed up learning. A many-to-one mapping for knowledge transfer in RL was presented in [11], which used a linear combination of relevant states and actions to initialize learning. However, it needed humans to provide the parameters of the linear combination, and the optimal values of the parameters were not easy to reach [12]. To avoid this, some researchers developed mechanisms for optimal mapping selection. Anestis and Ioannis [13] proposed two algorithms to select the optimal mapping for both model-based and model-free RLs respectively. In another of their works [14], they proposed a method which autonomously selected mappings from a set of all possible inter-task mappings.

In order to learn similarity of tasks without human intervention, some automatic works have been proposed. Ferns [15] measured the distance between Markov decision processes (MDPs) by constructing bisimulation metrics on rewards and probability functions. Taylor [16] gave a method called "modeling approximate state transitions by exploiting regression (MASTER)", which automatically learned a mapping from one task to another based on evidences gathered from task environments. A method using neural work to automatically map actions in one domain to actions in the other domain was proposed in [17]. Carroll and Seppi [18] defined similarity in terms of tasks, and proposed several possible task similarity measures based on the transfer time, policy overlap, Q-values, and reward structure respectively to measure the similarity between tasks. Teng [19] proposed to use self-organization neural networks to make the effective use of domain knowledge to reduce model complexity in RL. Through minimizing the reconstruction error of a restricted Boltzmann machine simulating the behavioral dynamics of two compared Markov decision processes, Ammar and Eaton [20] gave a data-driven automated Markov decision process similarity metric. Song and Gao [21] introduced two metrics for measuring the distance between finite Markov decision processes: The Hausdorff metric measuring the distance between two subsets of a metric space, and the Kantorovich metric measuring the distance between probabilistic distributions. These works are from a purely computational perspective, so they are computationally intensive, and cannot handle scenarios with a large number of source tasks [19].

To solve the issue raised when handling large number of tasks, recent works used the clustering to learn the similarity automatically. Li and Liao [23] used the intrinsic clustering property of the Dirichlet process to impose sharing of episodes among similar tasks, which effectively reduced the number of episodes required for learning a good policy in each task, when data sharing was appropriate. Lazaric and Ghavamzadeh [27] used a Gaussian process hierarchical BNP model to cluster different classes of source MDPs in sequential batch mode. Wilson and Fern [28] classified different Markov decision processes by clustering the dynamics and rewards which were drawn from the Dirichlet process

mixture models. A flexible Bayesian nonparametric model was exploited to cluster different classes of source Markov decision processes by the dependent Dirichlet process-based clusters of Gaussian process in [24]. However, these clustering-based methods need to use methods such as Bayesian model to simulate distributions, bias will arise during the simulation process, and the final results of the clustering-based methods are based on the simulated distributions, so the effects of these methods are easily affected by the simulation bias. Further, these methods suffer from the sensitivity to the noise and variation in the policies, dynamics, and value functions of source tasks.

We follow the automatic work line to select the highly-matched source tasks by LSM. LSM does not require the distribution model assumption for the task value space, thus reducing the bias from the assumption. Also, latent structure deriving in LSM benefits the suppression of noise and outliers. Furthermore, structural alignment of LSM guarantees the efficiency of handling a large number of tasks.

### 2.2. Low Rank Embedding

The algorithm we use to derive latent structures of value function space falls in the category of low rank embedding (LRE) [29]. Low rank embedding is a two-part algorithm for solving embedding in low-dimensional space by using the low rank matrix approximation method. Now given a data set $X$, at the beginning, LRE minimizes the following loss function so as to calculate the reconstruction coefficient matrix $\hat{X}$:

$$\frac{1}{2}\|X - X\hat{X}\|_F^2 + \lambda\|\hat{X}\|_*, \tag{1}$$

where $\lambda > 0$, $\|X\|_F = \sqrt{\sum_i \sum_j |X_{i,j}|^2}$ is the Frobenius norm. As for singular values $\sigma_i$, $\|X\|_* = \sum_i \sigma_i(X)$ is the spectral norm. LRE uses the alternating direction method of multipliers (ADMM) algorithm [30] to minimize (1) to calculate the reconstruction coefficient matrix $\hat{X}$, and then minimizes the reconstruction error of $X$ in low-dimensional space to obtain the embedding $F$. The reconstruction error function is as follows:

$$\frac{1}{2}\|F - F\hat{X}\|_F^2 s.t. F^T F = \mathbb{I}, \tag{2}$$

where $\mathbb{I}$ is the identity matrix, and the constraint $F^T F = \mathbb{I}$ can remove arbitrary scaling factors in the low-dimensional embedding and can ensure that the minimization problem of (2) is a well-posed problem. In [31], it is shown that (2) can be minimized to obtain the embedding $F$ by calculating the $d$ smallest non-zero eigenvectors of the Gram matrix $(\mathbb{I} - \hat{X})^T(\mathbb{I} - \hat{X})$.

To estimate efficiently the similarity between source and target tasks, we modify low rank embedding, and utilize the modified algorithm in RL (please see Section 3.1). To the best of our knowledge, our approach is a novel attempt to use LRE in knowledge transfer for RL.

## 3. Method

The proposed method is divided into two stages. Figure 1 shows an overview of the two-staged method composed of latent structure matching and value function transfer.

As shown in Figure 1, in the first stage, latent structures are derived to estimate the similarity between tasks and find the highly-matched source task for a new target task. In the second stage, value function transfer is conducted to improve the exploration strategy of agents by using action advice extracted from the highly-matched source task. The details of each stage will be given in the following subsections.

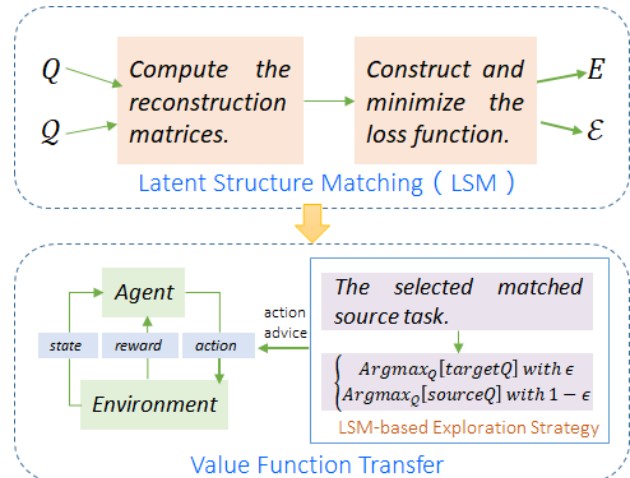

**Figure 1.** Overview of the two-staged method composed of latent structure matching and value function transfer. In the first stage, latent structure matching (LSM) is used to measure the similarity between tasks, so as to find the highly-matched source task for the target task. In the second stage, value function transfer is used to improve the agent's exploration strategy, thereby improving the agent's performance.

### 3.1. Latent Structure Matching (LSM)

The algorithmic basis of LSM is manifold alignment [32], through which LSM derives latent structures of tasks and finds matched source task for a new target task. LSM is actually one of the variants of LRE. Like normal LRE, it firstly calculates a reconstruction coefficient matrix for source and target tasks, and then minimizes the reconstruction error in low-dimensional space. However, compared with normal LRE, several modifications are made in the reconstruction error function and reconstruction coefficient matrix for efficient estimation of similarity between source and target tasks.

Given action value functions learned for $S$ source tasks, we arrange all these functions into a vector $Q = [Q_l^{(s)}]_{1 \times L}$, where $Q_l^{(s)}$ is the $l^{th}$ action value function for the $s^{th}$ source task. Similarly, all of the action value functions learned for $T$ target tasks are also arranged into the vector $\mathcal{Q} = [\mathcal{Q}_\gamma^{(t)}]_{1 \times \Gamma}$, where $\mathcal{Q}_\gamma^{(t)}$ is the $\gamma^{th}$ action value function for the $t^{th}$ target task.

The Equation (2) can be used to derive source and target tasks' latent structures, namely the low-dimensional embeddings of action value functions $Q$ and $\mathcal{Q}$. To match the embeddings in low dimension, we add an alignment item and obtain our error function:

$$S(\Phi) = (1 - \mu)\|\Phi - B\Phi\|_F^2 + \mu \sum_{i,j=1}^{m} \|\Phi_i - \Phi_j\|^2 C_{i,j}, \tag{3}$$

where $\mu \in [0, 1]$ is a weight parameter. The first item of (3) represents the reconstruction error, and it is consistent with the right side of (2); the second item is added to represent the alignment error. $\Phi \in \mathbb{R}^{m \times d}$, $B \in \mathbb{R}^{m \times m}$, and $C \in \mathbb{R}^{m \times m}$ are all block matrices:

$$\Phi = \begin{bmatrix} E \\ \mathcal{E} \end{bmatrix}, B = \begin{bmatrix} \hat{Q} & 0 \\ 0 & \hat{\mathcal{Q}} \end{bmatrix}, C = \begin{bmatrix} 0 & C^{(Q,\mathcal{Q})} \\ C^{(\mathcal{Q},Q)} & 0 \end{bmatrix},$$

where $m = L + \Gamma$ and $d$ is the dimension of low-dimensional space. $E$ and $\mathcal{E}$ are the embeddings respectively for $Q, \mathcal{Q}$. $C^{(Q,\mathcal{Q})}$ is the proximity matrix, which can be obtained by using cosine similarity metric. $\hat{Q}$ and $\hat{\mathcal{Q}}$ are the reconstruction coefficient matrices of $Q$ and $\mathcal{Q}$, respectively, which can be solved by minimizing (1). However, the minimization of (1) in LRE requires frequent iterative calculation. For this, we use the closed-form solution of subspace estimation [33] to calculate the

reconstruction coefficient matrix, thereby reducing the calculation. The specific procedure is described in Appendix A.

The error function in (3) can be reduced to a sum of matrix traces:

$$
\begin{aligned}
S(\Phi) &= (1-\mu)tr((\Phi - B\Phi)^T(\Phi - B\Phi)) + \mu \sum_{z=1}^{d} \sum_{i,j=1}^{m} \|\Phi_{i,z} - \Phi_{j,z}\|_2^2 C_{i,j} \\
&= (1-\mu)tr(((\mathbb{I}-B)\Phi)^T(\mathbb{I}-B)\Phi) + 2\mu \sum_{z=1}^{d} \Phi_{.,z}^T L \Phi_{.,z} \\
&= (1-\mu)tr(\Phi^T(\mathbb{I}-B)^T(\mathbb{I}-B)\Phi) + 2\mu tr(\Phi^T L\Phi).
\end{aligned}
\tag{4}
$$

Similar to locally linear embedding [34] and LRE, we introduce the constraint $\Phi^T\Phi = \mathbb{I}$ to ensure that the minimization problem of error function $S$ is a well-posed problem. Thus, we have

$$
argmin_{\Phi:\Phi^T\Phi=\mathbb{I}} S = argmin_{\Phi:\Phi^T\Phi=\mathbb{I}} (1-\mu)tr(\Phi^T A\Phi) + 2\mu tr(\Phi^T L\Phi),
\tag{5}
$$

where $A = (\mathbb{I}-B)^T(\mathbb{I}-B)$, and $L$ is the combinatorial graph Laplacian $L = D - C$, where $D$ is the diagonal matrix of row sums $D_{i,i} = \sum_j C_{i,j}$. To construct a loss function from (5), a Lagrange multiplier $\delta$ is introduced to the right side of (5):

$$
\mathcal{L}(\Phi,\delta) = (1-\mu)tr(\Phi^T A\Phi) + 2\mu tr(\Phi^T L\Phi) + \langle \delta, \Phi^T\Phi - \mathbb{I} \rangle.
\tag{6}
$$

To minimize the loss function (6), the roots for the partial derivatives of $\mathcal{L}$ for $\Phi$ and $\delta$ are calculated as:

$$
\begin{cases}
\frac{\partial \mathcal{L}}{\partial \Phi} = (1-\mu)A\Phi + 2\mu L\Phi - \delta\Phi = 0 \\
\frac{\partial \mathcal{L}}{\partial \delta} = \Phi^T\Phi - \mathbb{I} = 0.
\end{cases}
\tag{7}
$$

Therefore, to optimize (5) to obtain embedding $\Phi$, we need to calculate the $d$ smallest non-zero eigenvectors of matrix $((1-\mu)A + 2\mu L)$. After the embedding $\Phi$ is obtained, the distance between $E$ and $\mathcal{E}$, that represents the similarity between source and target task set, can be estimated. Finally, the highly-matched source task for a target task can be found. The whole procedure of LSM is fully described in pseudocode Algorithm 1.

---

**Algorithm 1** Latent structure matching

---

**Input:** source action value function $Q$; target action value function $\mathcal{Q}$; embedding dimension $d$; 
　　　weight parameter $\mu$
**Output:** embedding matrix $E$ of source task; embedding matrix $\mathcal{E}$ of target task.
　1: Compute the reconstruction coefficient matrices as described in Appendix A:
　2: Build the block matrices $\Phi$, $B$ and $C$.
　3: Construct the error function (3), namely

　　　$S(\Phi) = (1-\mu)\|\Phi - B\Phi\|_F^2 + \mu \sum_{i,j=1}^{m} \|\Phi_i - \Phi_j\|^2 C_{i,j}$

　4: Minimize the loss function (6):

　　　$\mathcal{L} = (1-\mu)tr(\Phi^T A\Phi) + 2\mu tr(\Phi^T L\Phi) + \langle \delta, \Phi^T\Phi - \mathbb{I} \rangle$

　5: The embedding matrices $E$ and $\mathcal{E}$ can be obtained by calculating the $d$ smallest non-zero 
　　　eigenvectors of matrix $(1-\mu)A + 2\mu L$.

---

### 3.2. Value Function Transfer

After finding the highly-matched source task, the learnt action value functions of the matched source task can be directly transferred to the target task. However, if there is not enough similarity

between source tasks and target tasks, it will result in the risk of negative transfer due to direct transfer. So using the action value functions of the matched source task to guide the agent in action exploration may be a safe program.

The most commonly used exploration strategy is the $\epsilon$-greedy exploration strategy [35], whose development purpose is to strike a balance between exploration and exploitation. This balance determines whether one should explore the invisible parts of the space or use the knowledge that has been acquired during the learning process. Specifically, the $\epsilon$-greedy exploration strategy performs exploitation with the probability $\epsilon$ by selecting a known action that maximizes the value function of a target task, and performs exploration with the probability 1-$\epsilon$ by randomly selecting an unknown action.

By using the knowledge from the source tasks, the $\epsilon$-greedy exploration strategy is modified as shown in Algorithm 2. The modified strategy, which we call an LSM-based exploration strategy, performs exploitation as normal $\epsilon$-greedy exploration strategy, but performs exploration with a probability of 1-$\epsilon$ by selecting the learnt action that maximizes the value function of the matched source task.

---

**Algorithm 2** LSM-based exploration strategy

---

**Input:** *targetQ*; *sourceQ*; $\epsilon$
**Output:** Action

$$Action \leftarrow \begin{cases} Argmax_a[targetQ], \text{with probability} \quad \epsilon \\ Argmax_a[sourceQ], \text{with probability} \ (1-\epsilon) \end{cases}$$

---

This exploratory action advice from LSM-based exploration strategy can reduce the instability from random exploration. Moreover, the modified exploration strategy can greatly speed up the learning of a target task, while achieving at least as good performance as $\epsilon$-greedy exploration strategy. The Equation (8) gives a definite proof for this assertion, and the specific proof details of (8) are given in Appendix B.:

$$Q_i^{\pi}(s,a) - Q_i^{\pi'}(s,a) \leq \varrho \, min_j \sum_{s'} p \sum_{a'} \omega \| Q_i^{\pi}(s',a') - Q_j^{\pi}(s',a') \|, \tag{8}$$

where $p$ denotes the probability $p(s'|s,a)$ in brief form, $w$ denotes the average probability $(\epsilon/\alpha)$ over the total number of actions $\alpha$. $\varrho$ is defined as $\gamma/[1-\gamma(1-\epsilon)]$ with the discount factor $0 < \gamma < 1$ . $Q$ denotes the action value function with its superscript denoting the used exploration strategy, and its subscript denoting the task index. $\pi$ and $\pi'$ denote respectively the $\epsilon$-greedy exploration strategy, and the LSM-based exploration strategy.

## 4. Experiments

In this section, we conduct several experiments on the maze navigation problem and the mountain car problem to verify the effectiveness of the proposed method. The experiments on maze navigation problem show how knowledge is transferred between similar environments with a tabular representation of discrete state and action space, and action value functions. The experiments on the mountain car problem illustrate a more complex transfer procedure under the continuous control. In this experiment, the radial basis function (RBF) network [36,37], an effective feed-forward neural network with the best approximation and global optimal performance, is utilized to estimate action value functions. Both experiments take Q-Learning framework [38], and the source codes for the experiments can be downloaded from the github link: https://github.com/yizhouzhouyi/LSM.git. The experimental results with the modified $\epsilon$-greedy exploration strategy are compared against the results from the traditional $\epsilon$-greedy exploration strategy.

### 4.1. Experiments on Maze Navigation Problem

These experiments are conducted on a 10×10 maze navigation environment in two-dimensional space, as shown in Figure 2. An agent is required to walk from the start point to the target point, while avoiding all obstacles in the path and maximizing the total rewards obtained.

The detailed settings about the maze navigation environments are listed below.

- *Action*: The action space is one-dimensional and discrete. The agent can perform four actions: Upward, downward, leftward, and rightward.
- *State*: The state space is also one-dimensional and discrete. Since each grid in the maze represents one state, there are a total of 100 states.
- *Reward*: When the agent reaches the target state, it will receive a reward of +10; when the agent encounters an obstacle, it will get a penalty of −10; and when the agent is in any other state, it will get a living penalty of −0.1.

We design two maze environments as shown in Figure 2. In two maze environments, the action value functions obtained from source tasks (the grids marked in blue in Figure 2a) and the action value functions from target tasks (the grids marked in yellow in Figure 2a,b) are used as the input of LSM.

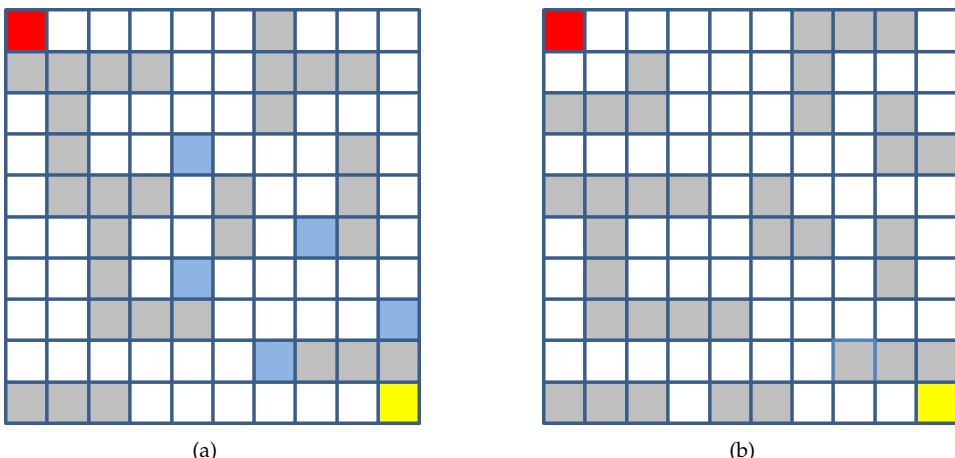

|       |       |
|:-----:|:-----:|
|  (a)  |  (b)  |

**Figure 2.** The maze navigation environments of two different settings. The red, yellow and gray grids in the figure denote respectively the starting positions, the target positions, and the obstacles. The goal of the agent is to walk from the start point (the red grids) to the target point (the yellow grids).

The experimental results are shown in Figure 3, the learning curves are smoothed with a window of 100 episodes with an error bar at every 100 episodes. The left curve shows the effect of knowledge transfer under the same environment settings, and the right curve shows the effect of knowledge transfer under different environment settings.

It can be seen from Figure 3 that LSM-based modified $\epsilon$-greedy exploration strategy performs significantly better than normal $\epsilon$-greedy exploration strategy. For two exploration strategies, the value of $\epsilon$ is set as 0.9.

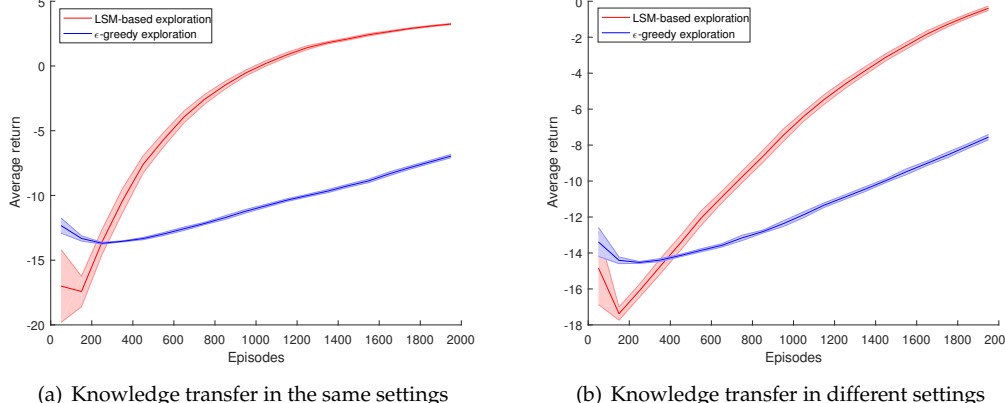

(a) Knowledge transfer in the same settings    (b) Knowledge transfer in different settings

**Figure 3.** The learning curves of LSM-based exploration strategy and $\epsilon$-greedy exploration strategy on the maze navigation problem. The curves are smoothed with a window of 100 episodes with an error bar at every 100 episodes.

To have an insight into the effect of $\epsilon$, we change the value of $\epsilon$ in the range $[0.5, 0.9]$ to observe the performance variation of two exploration strategies, and the results are given in Figures 4 and 5. The upper part of Figure 4 shows the average return of 2000 episodes obtained by changing the value of $\epsilon$ under the same environment settings, and the lower part of Figure 4 shows the average return of 2000 episodes obtained by changing the value of $\epsilon$ under different environment settings. As can be seen, compared with $\epsilon$-greedy exploration strategy, the LSM-based exploration strategy can always achieve a higher average return regardless of the value of $\epsilon$.

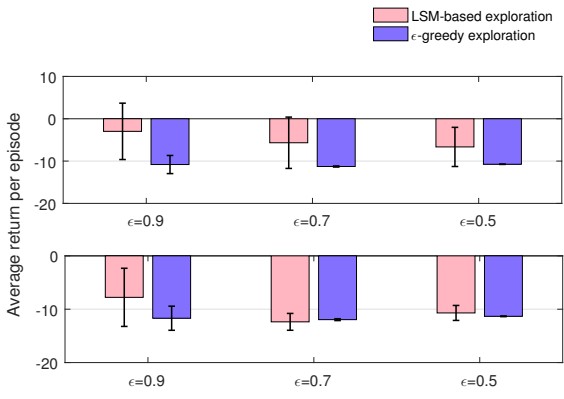

**Figure 4.** The comparison of the average return of 2000 episodes using the LSM-based and the $\epsilon$-greedy exploration strategies, and the values of $\epsilon$ are 0.9, 0.7, and 0.5 respectively from left to right. The upper part shows the results under the same environment settings and the lower part shows the results under different environment settings.

Figure 5 shows the variation in average return of each episode obtained by changing the value of $\epsilon$ under the same environment settings and different environment settings, respectively. When the value of $\epsilon$ is set to 0.7 or 0.5, the Q-Learning algorithm using $\epsilon$-greedy exploration strategy almost stops learning; however, the Q-Learning algorithm using LSM-based exploration strategy continues to perform well whether under the same environment settings or different environment settings. The reason for this may be that the increase of randomness in action selection greatly will reduce the possibility of taking sequent actions to find valid paths to target state; the good performance of the

modified strategy attributes to the knowledge transfer, which can reduce the randomness to some extent.

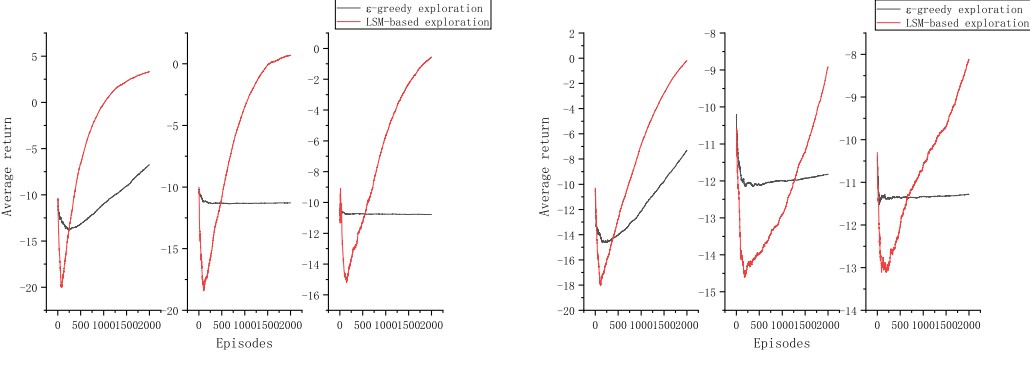

(a) Knowledge transfer in the same settings    (b) Knowledge transfer in different settings

**Figure 5.** The learning curves of LSM-based and $\epsilon$-greedy exploration strategies with different values of $\epsilon$ ( 0.9, 0.7, and 0.5 from left to right). The curves are smoothed with a window of 100 episodes with an error bar at every 100 episodes.

## 4.2. Experiments on Mountain Car Problem

These experiments are conducted on the mountain car [39,40] environment under the classical control problem of the OpenAI gym environment library. The mountain car problem belongs to the continuous control problem. As shown in Figure 6, in the experimental environment, a car is located on a two-dimensional track between two mountains; it is required to slide down from the left side of the mountain and rush to the target position on the right side of the mountain.

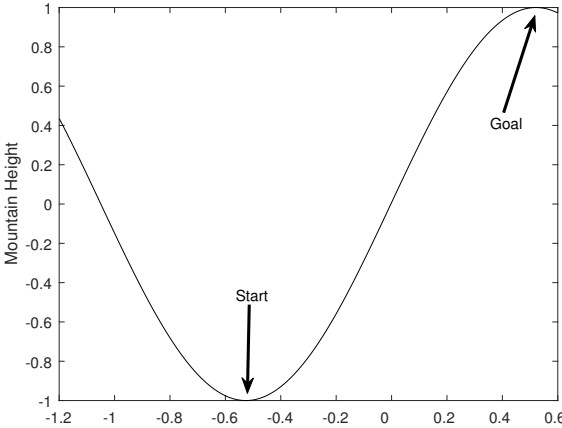

**Figure 6.** The mountain car environment. The purpose of the agent is to start from the starting position marked "Start", and use the energy obtained by sliding down from the left side of the mountain to rush to the position marked "Goal" on the right side of the mountain.

The detailed settings about the mountain car environment are listed below:

- *Action*: The action space is one-dimensional and discrete. The car can perform three actions, including leftward, rightward, and neutral, namely:

$$Action = (leftward, neutral, rightward).$$

- *Reward*: Each time the car arrives at a state, it obtains a reward of $-1.0$.

- *State*: The state space is two-dimensional and continuous. A state consists of speed and position, defined as:

$$\begin{cases} Velocity = (-0.007, 0.007) \\ Position = (-1.2, 0.6) \end{cases}$$

- *Initial state*: The position marked "Start" in Figure 6 denotes the initial state of the car. The initial state is set as:

$$\begin{cases} Position = -0.5 \\ Velocity = 0.0 \end{cases}$$

- *Goal state*: The position marked "Goal" in Figure 6 is the target state of the car. The target state is set as: $Position \geq 0.5$.
- *State update rule*: A state is updated as the rule:

$$\begin{cases} Velocity = Velocity + 0.001 \times (Action - 1) - 0.0025 \times cos(3 \times Position) \\ Position = Position + Velocity \end{cases}$$

As state values in these experiments are continuous, we use the RBF network to approximate action value functions instead of the tabular representation. The parameters of estimated action value functions of source and target tasks are the input in LSM, used to find the matched source tasks, from which knowledge is transferred to improve the $\epsilon$-greedy exploration strategy.

We compare the traditional $\epsilon$-greedy exploration strategy with our strategy, namely the LSM-based exploration strategy, and the experimental results are given in Figure 7. The learning curves are smoothed with a window of five episodes with an error bar at every five episodes. According to the evaluation criteria for knowledge transfer methods in reinforcement learning proposed by Taylor [3], our strategy performs better than the initial performance, called Jumpstart [3], and the final performance, called Asymptotic Performance [3], of the $\epsilon$-greedy exploration strategy. In the mountain car environment, because the agent using $\epsilon$-greedy exploration strategy explores randomly, it needs to try many times to know which speed it should take to get to the left side of the mountain so that it can use the energy of gliding to reach the right side of the mountain. Further, because of the experience gained in previous tasks, the agent using LSM-based exploration strategy can find a good speed at the beginning, and then uses the feedback from the environment to quickly find the optimal speed. Therefore, our method can achieve better performance at the beginning and the end.

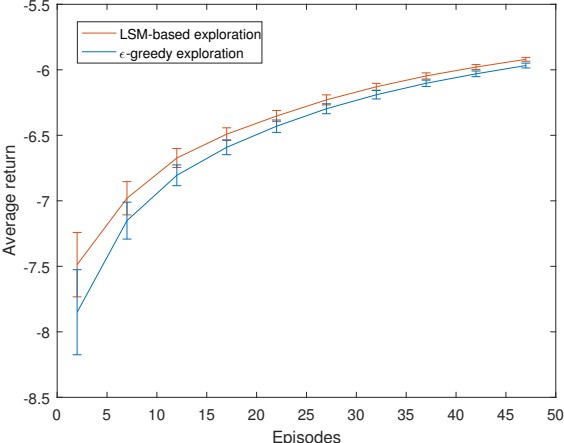

**Figure 7.** Learning curves for the mountain car problem. The curves are smoothed with a window of five episodes with an error bar at every five episodes.

## 5. Conclusions

In this paper, we propose a novel task matching algorithm to derive the latent structures of tasks, and align the structures to select highly-matched source tasks. We develop an improved exploration strategy to draw knowledge from the matched source tasks, in order to speed up the learning and to improve the performance of RL agents. We also give a theoretical proof to verify the improvement of the exploration strategy based on latent structure matching. Our method achieves significant performance improvement, when compared with the traditional $\epsilon$-greedy exploration strategy on the maze navigation problem and the mountain car problem. Such a method can prove to be useful for real-world applications such as autonomous robots, where sample efficiency is a critical factor. Further, in future work, causal inference may be introduced to transfer knowledge in reinforcement learning by adding causal factors.

**Author Contributions:** Y. Z. and F. Y. conceived and designed the theoretical calculations and experiments. Y. Z. and F. Y. carried out all the theoretical calculations. Y. Z. and F. Y. performed and analyzed the experiments. Both authors discussed the results and prepared and revised the manuscript.

**Funding:** This research was funded in part by National Nature Science Foundation of China (NSFC) under Grant 61371149.

**Acknowledgments:** This work is supported in part by National Nature Science Foundation of China (NSFC) under Grant 61371149.

**Conflicts of Interest:** The authors declare no conflict of interest.

## Appendix A

In this supplementary material, we describe the steps of the reconstruction coefficient matrix mentioned in Section 3.

First, Singular value decomposition (SVD) [41] is used to decompose $Q$ into the form $Q = UKV^T$, where $U$ is the left singular matrix having only a single element, $K = [k_i]_{1 \times L}$ is the singular value matrix, and $V = [v_{ij}]_{L \times L}$ is the right singular matrix. According to the value of elements in $K$, we divide $K$ into two submatrices, namely $K_1$ and $K_2$. Correspondingly, $V$ is also divided into two submatrices $V_1$ and $V_2$. The detailed division rule is defined as follows:

(a)  for $\forall k_i \in K$, if $k_i > 1$, it is put in the submatrix $K_1$; correspondingly, the $i^{th}$ column vector in $V$ is put in the submatrix $V_1$;

(b)  if $k_i \leq 1$, it is put in the submatrix $K_2$ ; correspondingly, the $i^{th}$ column vector in $V$ is put in the submatrix $V_2$.

Next, according to [33], the vector $Q$ is reconstructed as the matrix $\hat{Q} \in \mathbb{R}^{L \times L}$ by the following formula:

$$\hat{Q} = V_1 (\mathbb{I} - K_1^{-2}) V_1^T,$$

where $\mathbb{I}$ is the identity matrix. Using the same way, the reconstruction coefficient matrix $\hat{\mathcal{Q}} \in \mathbb{R}^{\Gamma \times \Gamma}$ of $\mathcal{Q}$ can be obtained.

## Appendix B

In this supplementary material, we give the proof details of (8) mentioned in Section 3. In the previous literature, two important theories about policy improvement have been proposed. One is Bellman's [42] policy improvement theory. This theory states that selecting actions greedily with respect to a policy's value function gives rise to another policy whose performance is no worse than the former's. Another one is Andre's [43] generalized policy improvement (GPI), which extends Bellman's policy improvement theory to the scenario where the new policy is to be computed based on value functions of a set of source tasks.

Here, according to GPI, we extend the notations to make it more convenient to depict the quantities involved in transfer learning. In this supplementary material, $j \in \{1, 2, \cdots, L\}$ and $i \in \{1, 2, \cdots, \Gamma\}$ denote a task from source task set $Y$ and target task set $T$, respectively. Moreover, we use $\pi$ to denote

the $\epsilon$-greedy exploration strategy, and $\pi'$ to denote the modified exploration strategy with knowledge transfer, namely LSM-based exploration strategy. $Q_j^\pi$ is used to denote the action value function with the policy $\pi$ on task $j$.

Suppose we have learnt action value functions $Q_i^\pi$ and $Q_i^{\pi'}$. To examine the performance improvement of LSM-based exploration strategy, we compare it with traditional $\epsilon$-greedy exploration strategy by calculating their difference. Based on the Bellman equation [42], we have

$$Q_i^\pi(s,a) - Q_i^{\pi'}(s,a)$$
$$= R_i(s,a) + \gamma \sum_{s'} p(s'|s,a) \sum_{a'} \pi(a'|s') Q_i^\pi(s',a') - R_i(s,a) - \gamma \sum_{s'} p(s'|s,a) \sum_{a'} \pi'(a'|s') Q_i^{\pi'}(s',a')$$
$$= \gamma \sum_{s'} p(s'|s,a) \sum_{a'} \pi(a'|s') Q_i^\pi(s',a') - \gamma \sum_{s'} p(s'|s,a) \sum_{a'} \pi'(a'|s') Q_i^{\pi'}(s',a')$$
$$= \gamma \sum_{s'} p(s'|s,a) [\sum_{a'} \pi(a'|s') Q_i^\pi(s',a') - \sum_{a'} \pi'(a'|s') Q_i^{\pi'}(s',a')]$$

Then, according to the exploration rules of the $\epsilon$-greedy exploration strategy and the LSM-based exploration strategy as mentioned above, the above equation is rewritten as

$$Q_i^\pi(s,a) - Q_i^{\pi'}(s,a)$$
$$= \gamma \sum_{s'} p(s'|s,a) [(1-\epsilon) max Q_i^\pi(s',a') + \epsilon/w \sum_{a'} Q_i^\pi(s',a') - (1-\epsilon) max Q_i^{\pi'}(s',a') - \epsilon max Q_j^\pi(s',a')]$$
$$= \gamma \sum_{s'} p(s'|s,a) [(1-\epsilon) max Q_i^\pi(s',a') - (1-\epsilon) max Q_i^{\pi'}(s',a')] + \gamma \sum_{s'} p(s'|s,a) [\epsilon/w \sum_{a'} Q_i^\pi(s',a') - \epsilon max Q_j^\pi(s',a')]$$
$$\leq \gamma \sum_{s'} p(s'|s,a) [(1-\epsilon) max Q_i^\pi(s',a') - (1-\epsilon) max Q_i^{\pi'}(s',a')] + \gamma \sum_{s'} p(s'|s,a) [\epsilon/w \sum_{a'} Q_i^\pi(s',a') - \epsilon/w \sum_{a'} Q_j^\pi(s',a')]$$
$$\leq \gamma \sum_{s'} p(s'|s,a)(1-\epsilon) max [Q_i^\pi(s',a') - Q_i^{\pi'}(s',a')] + \gamma \sum_{s'} p(s'|s,a) \sum_{a'} \epsilon/w [Q_i^\pi(s',a') - Q_j^\pi(s',a')]$$
$$\leq \gamma \sum_{s'} p(s'|s,a)(1-\epsilon) max [Q_i^\pi(s',a') - Q_i^{\pi'}(s',a')] + \gamma \sum_{s'} p(s'|s,a) \sum_{a'} \epsilon/w \mid Q_i^\pi(s',a') - Q_j^\pi(s',a') \mid$$
$$= \gamma(1-\epsilon)\Delta + \gamma \sum_{s'} p(s'|s,a) \sum_{a'} \epsilon/w \mid Q_i^\pi(s',a') - Q_j^\pi(s',a') \mid,$$

where $\Delta = max[Q_i^\pi(s',a') - Q_i^{\pi'}(s',a')]$, and $w$ represents the total number of actions.

Since all values of $[Q_i^\pi(s',a') - Q_i^{\pi'}(s',a')]$ satisfy the above formula, the maximum value of $[Q_i^\pi(s',a') - Q_i^{\pi'}(s',a')]$, that is, $\Delta$, also satisfies the above formula. Thus, we obtain

$$\Delta \leq \gamma(1-\epsilon)\Delta + \gamma \sum_{s'} p(s'|s,a) \sum_{a'} \epsilon/w \|Q_i^\pi(s',a') - Q_j^\pi(s',a')\|$$
$$\Rightarrow \Delta \leq P \sum_{s'} p(s'|s,a) \sum_{a'} \epsilon/w \|Q_i^\pi(s',a') - Q_j^\pi(s',a')\|$$
$$\Rightarrow Q_i^\pi(s,a) - Q_i^{\pi'}(s,a) \leq \Delta \leq P \sum_{s'} p(s'|s,a) \sum_{a'} \epsilon/w \|Q_i^\pi(s',a') - Q_j^\pi(s',a')\|,$$

where $P = \gamma/[1 - \gamma(1-\epsilon)]$. At this point, (8) is proved.

As we can see from (8), the loss $[Q_i^\pi(s',a') - Q_i^{\pi'}(s',a')]$ is upper-bounded by one term. This term can be seen as a multiple of the differences between $Q_i^\pi$ (the value function learned on the target task) and the closest $Q_j^\pi$ (the value function achieved from the source task). This formalizes the intuition that an agent should perform well on target task $i$ if it has solved a similar task before. In a word, if we can match the source task for the target task with enough similarity, and transfer the value function of the source task to the target task to guide a new exploration strategy based on the $\epsilon$-greedy exploration strategy, then this new strategy will perform no worse than the $\epsilon$-greedy exploration strategy.

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
