# Peer review of "Latent Structure Matching for Knowledge Transfer in Reinforcement Learning"

_futureinternet, doi:10.3390/fi12020036_

Round 1

Reviewer 1 Report

This paper proposed a task matching algorithm to derive the latent structures of value functions for estimation where the highly-matched source tasks are selected. The authors are kindly advised to consider the following comments and revision requests.

1. Extensive improvement in English and scientific writing would be essential, e.g., the word "task" has been repeated numerous times in the abstract. Please unify the English presentation, whether it is American English or British, whether it is present form, past form or you want to use "we".
2. Order the references please, e.g., [27] is not appearing in a correct order. 3. Section 2.2., 3.2., and the first paragraph of the introduction need- to be rewritten to maintain originality and present minimum similarity with the work of former scientists. 4. elaborate more on the novelty with respect to (and comparison) reference 3. The description of the code must be moved to supplementary materials instead of the abstract.

Reviewer 2 Report

The paper proposes an improved knowledge transfer approach for reinforcement learning. The topic is interesting and worth investigating.

The paper is well written. The literature review is almost adequate. The approach is described using enough details and is compared with a baseline approach.

# Recommendations

The introduction section should also mention a few areas in which reinforcement learning has been used successfully.

Line 14, the authors are advised not to include links in the abstract. The purpose of the abstract should briefly mention the problem that will be addressed in the paper, highlight the novelty of the proposed approach and the results that have been obtained. 

At line 42 the authors mention several papers [12-18]. Further details should be included for the references 12, 13, 14, 15, 16. The paper already discusses references 17 and 18.

Reviewer 3 Report

REVIEW: Latent Structure Matching for Knowledge Transfer in

Reinforcement Learning

In this article, the authors improve Reinforcement Learning (RL) introducing a transfer learning with a novel task matching algorithm, proposed to derive the latent structures of value functions of tasks.  Mountain Car experiment is conducted to illustrate the significant performance gain of the improved exploration strategy, compared with previous approaches, and also a theoretical proof is given.

After the review, I recommended  the manuscript for publication only when the main issues exposed are improved or corrected.

*** Main issues *** 

The work has a general interest that matches Future Internet objectives but, as a general comment to improve the article, there should be a greater effort to approach the paper to potential readers of the field of computer science in a broad sense and other related fields.

1.- Abstract.  In my opinion, authors should remove in lines 13-14 “(The source codes for the experiments can be downloaded from the github link: https://github.com/yizhouzhouyi/LSM.git )”. This is understood (it is usual in literature) and in any case they will be explained in the main text, not in the abstract.

2.- Keywords: I suggest to add: “Mountain Car”.

Introduction

3.- Although I am not a native speaker, there are some problems of expression and redundancies in English, which should be corrected in the text, starting with: (line 19) “Reinforcement learning (RL) learns…” We must avoid repeating what is defined in the definition (look for alternative to "learns" here).

4.- On the other hand, authors miss here some fundamental theoretical references, such as classical work (same authors of reference 1): R.S. Sutton, A.G. Barto (1998) Reinforcement Learning: An Introduction. MIT Press, Cambridge, MA., and more recently Bianchi et al(2015) https://doi.org/10.1016/j.artint.2015.05.008 (ref 3??) In general, I recommend that authors review the theoretical framework of this article to improve this introduction explaining more in depth concepts such as “heuristics” and “transfer learning” and quote properly in the main text to avoid plagiarism.

5.- In the latter (Bianchi et al, 2015), authors define: “Reinforcement learning (RL) is a field of machine learning whose aim is to maximise the total amount of reward an agent receives while interacting with its environment. This interaction occurs by means of exploring the state space by trial-and-error actions on the environment, leading to a process whose convergence is often slow (or infeasible) on complex tasks ”. The quotation is almost textual so it is recommended to quote properly this work and related quoted in it: I think it's your reference 3, but it's incomplete (authors are missing and the order is incorrect). Please review carefully references!

6.- Although it is not a bad explanation (in lines 35-40 and subsequently 41-50), for a better understanding of the theoretical framework, I suggest that the authors should make a figure/scheme or table with an overview of the automatic similarity estimation works, summarizing the main features of the main types of approaches they cite. This would help readers and scholars to better contextualize their contribution (LSM) and to understand better the concepts introduced in section 2, improving the paper.

7.- On the other hand, since they are going to focus on LSM, Introduction is the place to explain in detail the experiments conducted (simulated navigation environment and the Mountain Car environment). The article should be self-contained, that is, a reader should be able to understand these experiments, for which a brief explanation is essential.

Related work

8.-  This first subsection (2.1.) is somewhat conflicting: there are some repetitions regarding the Introduction (i.e.” Early works were semi-automatic, as they conducted the similarity estimation of tasks according to human experience or intuition., lines 80-81). Authors must decide which part of the theoretical framework is presented in the introduction, and which part is presented here. I advise you that the most general aspects and definitions go in the introduction, and that you really focus here on the most recent works, directly related to yours. Please rewrite consistently and without repetitions both sections.

9.- Please provide a general reference to justify (lines 113-114) “But these clustering based methods suffer from the bias from the modeling for clustering and the sensitivity to the noise and variation in the policies, dynamics, and value functions of source tasks.” or explain this bias better.

10.- Review syntax of formula and constraint in equation 2.

11.- Although in a next field, please review https://arxiv.org/abs/1712.01727 and references therein to be sure about that this is the first time to use LRE in knowledge transfer for RL.

Method

12.- I like Figure 1. It is the kind of graphic effort that helps to follow the article and that I have asked you for the theoretical introduction. However, the authors should explain it a bit more in the figure caption.

13.- In line 186 authors add: “The proof details of (8) are given in Appendix B.”. please place this sentence in line 181 to clarify that is not proven here (The proof details of (8) are given in Appendix B).

Experiments.

14.- Please, clarify that the experiments are explained in each subsection with more detail here at the very beginning (or in the introduction), and in the figure captions accordingly.

15.- Figure 4. Please, explain better in the caption this results.

16.- Lines 250-254: please explain in more detail why your strategy is better than the traditional e-greedy exploration strategy, quoting the classical references.

17.- Include in this section properly the comment from the abstract: “The source codes for the experiments can be downloaded from the github link: https://github.com/yizhouzhouyi/LSM.git”

Conclusion

18.- Finally, I think the study is interesting enough to expect the authors to be somewhat more daring in the conclusion, extremely short. I mean, it is important to go beyond mere (important) calculations, and consider how their results can improve other related fields and contrast with previous works, i.e. in heuristics (Bianchi et al(2009) https://doi.org/10.1007/978-3-642-02998-1_7 ) or probabilistic policy reuse (Fernández & Veloso (2006) https://dl.acm.org/doi/abs/10.1145/1160633.1160762 )

If not all of them, I think the authors should try to speculate a bit in the very short conclusion about the answers at least of some of my questions, and explain future lines of research or applications.

References

19.- Please review it carefully, i.e. ref [3].
